# Quantum Dynamics in a Fluctuating Environment

**DOI:** 10.3390/e21111040

**Published:** 2019-10-25

**Authors:** Xiangji Cai

**Affiliations:** School of Science, Shandong Jianzhu University, Jinan 250101, China; xiangjicai@foxmail.com

**Keywords:** quantum dynamics, open quantum systems, environmental fluctuations

## Abstract

We theoretically investigate the dynamics of a quantum system which is coupled to a fluctuating environment based on the framework of Kubo-Anderson spectral diffusion. By employing the projection operator technique, we derive two types of dynamical equations, namely, time-convolution and time-convolutionless quantum master equations, respectively. We derive the exact quantum master equations of a qubit system with both diagonal splitting and tunneling coupling when the environmental noise is subject to a random telegraph process and a Ornstein-Uhlenbeck process, respectively. For the pure decoherence case with no tunneling coupling, the expressions of the decoherence factor we obtained are consistent with the well-known existing ones. The results are significant to quantum information processing and helpful for further understanding the quantum dynamics of open quantum systems.

## 1. Introduction

A quantum system loses coherence information in the dynamical evolution resulting from the unavoidable environmental coupling. Understanding the dynamics of open quantum systems can help us know the essence of decoherence and the reason for the transition from quantum to classical [1,2,3,4,5,6]. How to effectively obtain the exact quantum master equation for an open quantum system is an extremely challenging but very meaningful problem, which helps us further understand the real dynamical evolution of the system. During the last few decades, people have generally described the dynamics of open quantum systems within Markov approximation [7,8]. Recently, it has increasingly drawn much attention to study the dynamical evolution of open quantum systems with methods beyond Markov approximation in the community of quantum information science, ranging from quantum computing to quantum measurements [9,10,11,12,13,14,15,16,17].

In general, the environmental effects on open quantum systems can be dealt with by stochastic noise processes within the framework of classical and quantum treatments [18,19,20,21,22,23,24,25,26,27,28,29,30,31,32,33,34,35,36,37,38]. For example, Ornstein-Uhlenbeck noise (OUN), as an important Gaussian stochastic process, has been generally used to model the environmental effects in a large number of quantum mechanical systems which exhibit Gaussian fluctuations, such as, Berry phase, dynamical decoherence and construction of robust quantum gates induced by classical fluctuating environments [39,40,41,42]. Random telegraph noise (RTN), as an important non-Gaussian stochastic process, has been generally used to model the environmental effects in various quantum mechanical systems, such as, dynamical decoherence induced by low-frequency noise and fluorescence spectra of single molecules [43,44,45,46,47,48,49,50,51,52,53,54]. Investigation of the quantum non-Markovian dynamics induced by the effects of the environmental noise resulting from classical fluctuating environments is important to understand the environmental backaction of coherence and useful for further potential applications in the manipulation and control of quantum coherence.

In this paper, we theoretically investigate the quantum dynamics of a qubit system with both diagonal splitting and tunneling coupling coupled to a fluctuating environment based on spectral diffusion model initiated by Kubo and Anderson. We derive two types of quantum master equations, namely, time-convolution (TC) and time-convolutionless (TCL) quantum master equations by employing the projection operator technique. The TC and TCL quantum master equations in the second order expansion give the exact dynamical evolution of the qubit system induced by the RTN and OUN, respectively. When there is no tunneling coupling, the qubit system undergoes pure decoherence and we obtain the expressions of the decoherence factor which are consistent with the well-known existing ones.

This paper is organized as follows. In Section 2 we introduce the stochastic model within the framework initiated by Kubo and Anderson and derive the TC and TCL equations for the reduced density matrix of the quantum system by employing the projection operator technique. In Section 3, we study the quantum dynamics of a qubit system induced by the RTN and OUN, respectively. In Section 4 we give the concluding remarks of the present study.

## 2. Theoretical Framework

We consider a quantum system which interacts with a fluctuating environment based on the framework initiated by Kubo and Anderson. The environmental effects on the system are described by means of stochastic Hamiltonian of the quantum system as [18,19]
(1)$$H\left(t\right)=H_{0}+\delta H\left(t\right),$$
where $H_{0}$ is the unperturbed Hamiltonian of the system and δH(t) is caused by the environmental fluctuations which yields a stochastic noise process.

The time evolution of the stochastic density matrix satisfies the Liouville-von Neumann equation
(2)$$\frac{\partial }{\partial t}\rho \left(t;\delta \left(t\right)\right)=\left[L_{0}+L_{\delta }\left(t\right)\right]\rho \left(t;\delta \left(t\right)\right),$$
where the notation ρ(t;δ(t)) indicates the dependence of the stochastic fluctuation term δH(t) and we have used the super-operators for $L_{0}\left(\cdot \right)=-\frac{i}{\hbar }\left[H_{0},\left(\cdot \right)\right]$ and $L_{\delta }\left(t\right)\left(\cdot \right)=-\frac{i}{\hbar }\left[\delta H\left(t\right),\left(\cdot \right)\right]$. By taking the average over different realizations of the environmental fluctuations, wee can derive the reduced density matrix as ρ(t)=〈ρ(t;δ(t))〉.

In order to obtain the dynamical evolution of the reduced density matrix, we first transform Equation (Equation 2) into the interaction picture
(3)$$\frac{\partial }{\partial t}\rho^{I}\left(t;\delta \left(t\right)\right)=L_{\delta }^{I}\left(t\right)\rho^{I}\left(t;\delta \left(t\right)\right),$$
where we have defined $\rho^{I}\left(t;\delta \left(t\right)\right)=e^{-L_{0}t}\hat{\rho }\left(t;\delta \left(t\right)\right)$ and $L_{\delta }^{I}\left(t\right)=e^{-L_{0}t}L_{\delta }\left(t\right)e^{L_{0}t}$.

We define the projection operator P [1,20]
(4)$$P\rho^{I}\left(t;\delta \left(t\right)\right)=\langle \rho^{I}\left(t;\delta \left(t\right)\right)\rangle \equiv \rho^{I}\left(t\right).$$
and the complementary projector Q=I−P
(5)$$Q\rho^{I}\left(t;\delta \left(t\right)\right)=\rho^{I}\left(t;\delta \left(t\right)\right)-\langle \rho^{I}\left(t;\delta \left(t\right)\right)\rangle \equiv \rho^{I}\left(t;\delta \left(t\right)\right)-\rho^{I}\left(t\right),$$
with the properties of the projectors $P^{2}=P$, $Q^{2}=Q$ and PQ=QP=0. By performing the projection operators on Equation (Equation 3), we obtain the evolution
(6a)$$\begin{array}{cc} \frac{\partial }{\partial t}P\rho^{I}\left(t;\delta \left(t\right)\right)= & P\frac{\partial }{\partial t}\rho^{I}\left(t;\delta \left(t\right)\right)=PL_{\delta }^{I}\left(t\right)\rho^{I}\left(t;\delta \left(t\right)\right), \end{array}$$
(6b)$$\begin{array}{cc} \frac{\partial }{\partial t}Q\rho^{I}\left(t;\delta \left(t\right)\right)= & Q\frac{\partial }{\partial t}\rho^{I}\left(t;\delta \left(t\right)\right)=QL_{\delta }^{I}\left(t\right)\rho^{I}\left(t;\delta \left(t\right)\right). \end{array}$$

In terms of the identity I=P+Q, we can rewrite Equation (6) as
(7a)$$\begin{array}{cc} \frac{\partial }{\partial t}P\rho^{I}\left(t;\delta \left(t\right)\right)= & PL_{\delta }^{I}\left(t\right)P\rho^{I}\left(t;\delta \left(t\right)\right)+PL_{\delta }^{I}\left(t\right)Q\rho^{I}\left(t;\delta \left(t\right)\right), \end{array}$$
(7b)$$\begin{array}{cc} \frac{\partial }{\partial t}Q\rho^{I}\left(t;\delta \left(t\right)\right)= & QL_{\delta }^{I}\left(t\right)P\rho^{I}\left(t;\delta \left(t\right)\right)+QL_{\delta }^{I}\left(t\right)Q\rho^{I}\left(t;\delta \left(t\right)\right). \end{array}$$ The solution for $Q\rho^{I}\left(t;\delta \left(t\right)\right)$ in Equation (7b) can be expressed as
(8)$$Q\rho^{I}\left(t;\delta \left(t\right)\right)=\int_{0}^{t}dt^{\prime }g\left(t,t^{\prime }\right)QL_{\delta }^{I}\left(t^{\prime }\right)P\rho^{I}\left(t^{\prime }|\xi \left(t^{\prime }\right)\right)+g\left(t,0\right)Q\rho^{I}\left(0;\delta \left(0\right)\right),$$
where $g(t,t^{\prime })$ is the forward propagator which can be written as
(9)$$g\left(t,t^{\prime }\right)=T_{\leftarrow }\exp\left[\int_{t^{\prime }}^{t}d\tau QL_{\delta }^{I}\left(\tau \right)\right],$$
with the chronological time-ordering operator $T_{\leftarrow }$. The forward propagator $g(t,t^{\prime })$ satisfies the evolution
(10)$$\frac{\partial }{\partial t}g\left(t,t^{\prime }\right)=QL_{\delta }^{I}\left(t\right)g\left(t,t^{\prime }\right),$$
initially with the condition $g(t^{\prime },t^{\prime })=I$.

By substituting the solution for $Q\rho^{I}\left(t;\delta \left(t\right)\right)$ back into Equation ([Disp-formula FD7a-entropy-21-01040]), we obtain the evolution
(11)$$\frac{\partial }{\partial t}P\rho^{I}\left(t;\delta \left(t\right)\right)=PL_{\delta }^{I}\left(t\right)P\rho^{I}\left(t;\delta \left(t\right)\right)+\int_{0}^{t}dt^{\prime }PL_{\delta }^{I}\left(t\right)g\left(t,t^{\prime }\right)QL_{\delta }^{I}\left(t^{\prime }\right)P\rho^{I}\left(t^{\prime };\delta \left(t^{\prime }\right)\right)+I_{\mathrm{TC}}\left(t\right),$$
with the inhomogeneous operator induced by the initial environmental correlation
(12)$$I_{\mathrm{TC}}\left(t\right)=PL_{\delta }^{I}\left(t\right)g\left(t,0\right)Q\rho^{I}\left(0;\delta \left(0\right)\right).$$ If the quantum system and environment are initially uncorrelated, the initial state satisfies $P\rho^{I}\left(0;\delta \left(0\right)\right)=\rho^{I}\left(0\right)=\rho^{I}\left(0;\delta \left(0\right)\right)$ and $Q\rho^{I}\left(0;\delta \left(0\right)\right)=0$ and the third term $I_{\mathrm{TC}}\left(t\right)$ on the right-hand side of Equation (Equation 11) vanishes.

Expanding the forward propagator $g(t,t^{\prime })$ in Dyson series $g\left(t,t^{\prime }\right)=1+\sum_{n=1}^{\infty }g_{n}\left(t,t^{\prime }\right)$ with $g_{n}\left(t,t^{\prime }\right)=\int_{t^{\prime }}^{t}dt_{1}\cdots \int_{t^{\prime }}^{t_{n-1}}dt_{n}QL_{\delta }^{I}\left(t_{1}\right)\cdot \cdot \cdot QL_{\delta }^{I}\left(t_{n}\right)$ and in terms of the definitions in Equations (Equation 4) and (Equation 5), we obtain the TC master equation for the time evolution of the quantum system
(13)$$\frac{d}{dt}\rho^{I}\left(t\right)=\langle L_{\delta }^{I}\left(t\right)\rangle \rho^{I}\left(t\right)+\int_{0}^{t}dt^{\prime }K\left(t,t^{\prime }\right)\rho^{I}\left(t^{\prime }\right)+I_{\mathrm{TC}}\left(t\right),$$
where the time non-local operator satisfies
(14)$$\int_{0}^{t}dt^{\prime }K\left(t,t^{\prime }\right)\rho^{I}\left(t^{\prime }\right)=\sum_{n=2}^{\infty }\int_{0}^{t}dt_{1}\cdots \int_{0}^{t_{n-2}}dt_{n-1}{\langle L_{\delta }^{I}\left(t\right)L_{\delta }^{I}\left(t_{1}\right)\cdots L_{\delta }^{I}\left(t_{n-1}\right)\rangle }_{\mathrm{pc}}\rho^{I}\left(t_{n-1}\right),$$
in terms of the partial cumulants
(15)$${\langle L_{\delta }^{I}\left(t\right)L_{\delta }^{I}\left(t_{1}\right)\cdots L_{\delta }^{I}\left(t_{n-1}\right)\rangle }_{\mathrm{pc}}=\sum {\left(-1\right)}^{q-1}\prod \langle L_{\delta }^{I}\left(t\right)\cdots \rangle \langle L_{\delta }^{I}\left(t_{j}\right)\cdots \rangle \cdots ,$$
with *q* denoting the quantity of averages in the term and according to the time chronological order $t>t_{1}>\cdot \cdot \cdot >t_{n}$.

We further derive the TCL master equation for the dynamical evolution of the quantum system. On the right-hand side of Equation (Equation 8), we replace the expression of the stochastic density matrix by
(16)$$\rho^{I}\left(t^{\prime };\delta \left(t^{\prime }\right)\right)=G\left(t,t^{\prime }\right)\rho^{I}\left(t;\delta \left(t\right)\right),$$
where $G(t,t^{\prime })$ is the backward propagator expressed as
(17)$$G\left(t,t^{\prime }\right)=T_{\to }\exp\left[-\int_{t^{\prime }}^{t}L_{\delta }^{I}\left(\tau \right)d\tau \right],$$
with $T_{\to }$ indicating the antichronological time-ordering, we can express the solution for $Q\rho^{I}\left(t;\delta \left(t\right)\right)$ in Equation (Equation 8) as
(18)$$Q\rho^{I}\left(t;\delta \left(t\right)\right)=\int_{0}^{t}dt^{\prime }g\left(t,t^{\prime }\right)QL_{\delta }^{I}\left(t^{\prime }\right)PG\left(t,t^{\prime }\right)\left(P+Q\right)\rho^{I}\left(t;\delta \left(t\right)\right)+g\left(t,0\right)Q\rho^{I}\left(0;\delta \left(0\right)\right).$$

Introducing the super-operator
(19)$$\Sigma \left(t\right)=\int_{0}^{t}dt^{\prime }g\left(t,t^{\prime }\right)QL_{\delta }^{I}\left(t^{\prime }\right)PG\left(t,t^{\prime }\right),$$
the solution for $Q\rho^{I}\left(t;\delta \left(t\right)\right)$ can be reexpressed as
(20)$$Q\rho^{I}\left(t;\delta \left(t\right)\right)={\left[1-\Sigma (t)\right]}^{-1}\Sigma \left(t\right)P\rho^{I}\left(t;\delta \left(t\right)\right)+{\left[1-\Sigma (t)\right]}^{-1}g\left(t,0\right)Q\rho^{I}\left(0;\delta \left(0\right)\right).$$ Substituting the solution for $Q\rho^{I}\left(t;\delta \left(t\right)\right)$ in Equation (Equation 20) back into Equation ([Disp-formula FD7a-entropy-21-01040]) gives the evolution
(21)$$\frac{\partial }{\partial t}P\rho^{I}\left(t;\delta \left(t\right)\right)=PL_{\delta }^{I}\left(t\right){\left[1-\Sigma \left(t\right)\right]}^{-1}P\rho^{I}\left(t;\delta \left(t\right)\right)+I_{\mathrm{TCL}}\left(t\right),$$
with the inhomogeneous operator caused by the initial environmental correlation
(22)$$I_{\mathrm{TCL}}\left(t\right)=PL_{\delta }^{I}\left(t\right){\left[1-\Sigma (t)\right]}^{-1}g\left(t,0\right)Q\rho^{I}\left(0;\delta \left(0\right)\right).$$ The second term $I_{\mathrm{TCL}}\left(t\right)$ on the right-hand side of Equation (Equation 21) vanishes if the quantum system and environment are initially uncorrelated due to the fact that $P\rho^{I}\left(0;\delta \left(0\right)\right)=\rho^{I}\left(0\right)=\rho^{I}\left(0;\delta \left(0\right)\right)$ and $Q\rho^{I}\left(0;\delta \left(0\right)\right)=0$.

Expanding the super-operator Σ(t) in series $\Sigma \left(t\right)=\sum_{n=1}^{\infty }\Sigma_{n}\left(t\right)$ and in terms of the definitions in Equations (Equation 4) and (Equation 5), we obtain the TCL master equation for the dynamical evolution of the quantum system
(23)$$\frac{d}{dt}\rho^{I}\left(t\right)=K\left(t\right)\rho^{I}\left(t\right)+I_{\mathrm{TCL}}\left(t\right),$$
where the time-local operator can be expressed as
(24)$$K\left(t\right)=\sum_{n=1}^{\infty }\int_{0}^{t}dt_{1}\cdots \int_{0}^{t_{n-2}}dt_{n-1}{\langle L_{\delta }^{I}\left(t\right)L_{\delta }^{I}\left(t_{1}\right)\cdots L_{\delta }^{I}\left(t_{n-1}\right)\rangle }_{\mathrm{oc}},$$
based on the time-order cumulants
(25)$${\langle L_{\delta }^{I}\left(t\right)L_{\delta }^{I}\left(t_{1}\right)\cdots L_{\delta }^{I}\left(t_{n-1}\right)\rangle }_{\mathrm{oc}}=\sum {\left(-1\right)}^{q-1}\prod \langle L_{\delta }^{I}\left(t\right)\cdots L_{\delta }^{I}\left(t_{i}\right)\rangle \langle L_{\delta }^{I}\left(t_{j}\right)\cdots L_{\delta }^{I}\left(t_{k}\right)\rangle \cdots .$$
with the sum taken over all possible divisions by keeping the time chronological order $t>\cdot \cdot \cdot >t_{i},t_{j}>\cdot \cdot \cdot >t_{k}$ and so on.

We have formally derived above two types of quantum master equations for dynamical evolution of the reduced density matrix of the system by employing the projection operator technique which are closely associated with the statistical properties of the environmental noise ξ(t). However, only for a few simple cases, we can obtain the exact expression of the reduced density matrix of the quantum system based on the two types of quantum master equations we have derived. In most cases, we need to take some approximations to obtain the reduced density matrix of the system or we should know the closure of the higher-order correlation functions of the environmental noise ξ(t) [33,36]. In the following section, we will study two special models of which the reduced density matrix can be exact solved.

## 3. Application and Discussion

We consider a two state qubit system with the intrinsic Hamiltonian
(26)$$H_{0}=\frac{\hbar }{2}\left(\omega_{0}\sigma_{z}+\Delta_{0}\sigma_{x}\right),$$
where $\sigma_{x,z}$ represent the Pauli matrices and $\omega_{0}$ and $\Delta_{0}$ denote the diagonal splitting and tunneling coupling between the states |1〉 and |0〉, respectively, In principle, the environmental effect gives rise to the stochastic fluctuations in both longitudinal and transverse directions in the presence of a fluctuating environment. We focus, here and in the following, mainly on the case that fluctuates longitudinally in diagonal splitting with the stochastic fluctuations
(27)$$\delta H\left(t\right)=\frac{\hbar }{2}\xi \left(t\right)\sigma_{z},$$
where ξ(t) denotes the environmental noise subject to a stochastic process. The case of transverse fluctuations in tunneling coupling can be dealt with in a similar way. For simplify, we make the assumption that the quantum system and environment are uncorrelated initially.

### 3.1. Quantum Dynamics Induced by the RTN Process

We first consider the dynamical evolution of the qubit system when the environmental noise ξ(t) yields a stationary RTN process of which the amplitude jumps between the two values ±ν randomly with an constant switching rate λ. The time evolution of the conditional probability for the RTN process obeys the master equation
(28)$$\begin{array}{cc} \frac{\partial }{\partial t}P\left(\nu ,t|\xi^{\prime },t^{\prime }\right)= & -\lambda P\left(\nu ,t|\xi^{\prime },t^{\prime }\right)+\lambda P\left(-\nu ,t|\xi^{\prime },t^{\prime }\right),\\ \frac{\partial }{\partial t}P\left(-\nu ,t|\xi^{\prime },t^{\prime }\right)= & -\lambda P\left(-\nu ,t|\xi^{\prime },t^{\prime }\right)+\lambda P\left(\nu ,t|\xi^{\prime },t^{\prime }\right), \end{array}$$
where the initial condition is given by $P\left(\xi ,t^{\prime }|\xi^{\prime },t^{\prime }\right)=\delta_{\xi ,\xi^{\prime }}$ for ξ=±ν. The solution of the conditional probability in Equation (Equation 28) can be written as
(29)$$P\left(\xi ,t|\xi^{\prime },t^{\prime }\right)=\frac{1}{2}\left[1+e^{-2\lambda (t-t^{\prime })}\right]\delta_{\xi ,\xi^{\prime }}+\frac{1}{2}\left[1-e^{-2\lambda (t-t^{\prime })}\right]\delta_{-\xi ,\xi^{\prime }}.$$ For positive λ and large time difference $\lambda (t-t^{\prime })\gg 1$, we obtain the stationary distribution
(30)$$P_{\mathrm{st}}\left(\xi \right)=\frac{1}{2}\left(\delta_{\xi ,\nu }+\delta_{\xi ,-\nu }\right).$$ Considering the noise process is Markov, the *n*-point probability distribution can be expressed as
(31)$$P\left(\xi_{n},t_{n};\cdots ;\xi_{1},t_{1}\right)=\prod_{i}^{n-1}P\left(\xi_{i+1},t_{i+1}|\xi_{i},t_{i}\right)P_{\mathrm{st}}\left(\xi_{1}\right).$$

Based on the stationary and Markovian statistical properties, the average of the noise process is zero
(32)〈ξ(t)〉=0,
and its second-order correlation function is exponential
(33)$$\langle \xi \left(t\right)\xi \left(t^{\prime }\right)\rangle =\nu^{2}e^{-2\lambda (t-t^{\prime })}.$$ Its higher order correlation functions satisfy the factorization relation [55,56]
(34)$$\langle \xi \left(t\right)\xi \left(t_{1}\right)\cdots \xi \left(t_{n}\right)\rangle =\langle \xi \left(t\right)\xi \left(t_{1}\right)\rangle \langle \xi \left(t_{2}\right)\cdots \xi \left(t_{n}\right)\rangle ,$$
for all sets of the time sequences with $t>t_{1}>\cdots >t_{n}\left(n\geq 2\right)$. We can, based on the statistical characteristics of the environmental noise ξ(t) obtained above, derive that its partial cumulants beyond second order are zero
(35)$${\langle \xi \left(t\right)\xi \left(t_{1}\right)\cdots \xi \left(t_{n}\right)\rangle }_{\mathrm{pc}}=0.$$

As a consequence, we can use the TC master equation to describe the time evolution for the reduced density matrix of the quantum system as [36]
(36)$$\begin{array}{cc} \frac{d}{dt}\rho \left(t\right)= & -\frac{i}{2}\left(\omega_{0}L_{\sigma_{z}}+\Delta_{0}L_{\sigma_{x}}\right)\rho \left(t\right)-\frac{1}{4}\int_{0}^{t}dt^{\prime }\langle \xi \left(t\right)\xi \left(t^{\prime }\right)\rangle \\ \; & \times L_{\sigma_{z}}\exp\left[-\frac{i}{2}\left(\omega_{0}L_{\sigma_{z}}+\Delta_{0}L_{\sigma_{x}}\right)\left(t-t^{\prime }\right)\right]L_{\sigma_{z}}\rho \left(t^{\prime }\right), \end{array}$$
where we use the definitions $L_{\sigma_{z}}\left(\cdot \right)=\left[\sigma_{z},\left(\cdot \right)\right]$ and $L_{\sigma_{x}}\left(\cdot \right)=\left[\sigma_{x},\left(\cdot \right)\right]$. The dynamical evolution of the reduced density matrix elements can be written as
(37)$$\begin{array}{cc} \frac{d}{dt}\rho_{11}\left(t\right)= & \frac{i}{2}\Delta_{0}\left[\rho_{10}\left(t\right)-\rho_{01}\left(t\right)\right],\\ \frac{d}{dt}\rho_{00}\left(t\right)= & -\frac{i}{2}\Delta_{0}\left[\rho_{10}\left(t\right)-\rho_{01}\left(t\right)\right],\\ \frac{d}{dt}\rho_{10}\left(t\right)= & -i\omega_{0}\rho_{10}\left(t\right)-\nu^{2}\int_{0}^{t}e^{-2\lambda (t-t^{\prime })}\left[\frac{\Delta_{0}^{2}}{2\Omega^{2}}+\frac{\Omega^{2}+\omega_{0}^{2}}{2\Omega^{2}}\cos\Omega \left(t-t^{\prime }\right)-i\frac{\omega_{0}}{\Omega }\sin\Omega \left(t-t^{\prime }\right)\right]\rho_{10}\left(t^{\prime }\right)\\ & +\frac{\nu^{2}\Delta_{0}^{2}}{2\Omega^{2}}\int_{0}^{t}e^{-2\lambda (t-t^{\prime })}\left[1-\cos\Omega \left(t-t^{\prime }\right)\right]\rho_{01}\left(t^{\prime }\right)+\frac{i}{2}\Delta_{0}\left[\rho_{11}\left(t\right)-\rho_{00}\left(t\right)\right],\\ \frac{d}{dt}\rho_{01}\left(t\right)= & i\omega_{0}\rho_{01}\left(t\right)-\nu^{2}\int_{0}^{t}e^{-2\lambda (t-t^{\prime })}\left[\frac{\Delta_{0}^{2}}{2\Omega^{2}}+\frac{\Omega^{2}+\omega_{0}^{2}}{2\Omega^{2}}\cos\Omega \left(t-t^{\prime }\right)+i\frac{\omega_{0}}{\Omega }\sin\Omega \left(t-t^{\prime }\right)\right]\rho_{01}\left(t^{\prime }\right)\\ & +\frac{\nu^{2}\Delta_{0}^{2}}{2\Omega^{2}}\int_{0}^{t}e^{-2\lambda (t-t^{\prime })}\left[1-\cos\Omega \left(t-t^{\prime }\right)\right]\rho_{10}\left(t^{\prime }\right)-\frac{i}{2}\Delta_{0}\left[\rho_{11}\left(t\right)-\rho_{00}\left(t\right)\right], \end{array}$$
where $\Omega =\sqrt{\omega_{0}^{2}+\Delta_{0}^{2}}$ denotes the eigen-splitting of the qubit system. We can take the Laplace transform over Equation (Equation 37) to obtain the solution of the reduced density matrix elements.

It is worth mentioning the uncoupled case $\Delta_{0}=0$ and the system undergoes pure decoherence. For this case, the time evolution of the reduced density matrix elements in Equation (Equation 37) can be reduced to
(38)$$\begin{array}{cc} \frac{d}{dt}\rho_{11}\left(t\right)= & 0,\\ \frac{d}{dt}\rho_{00}\left(t\right)= & 0,\\ \frac{d}{dt}\rho_{10}\left(t\right)= & -i\omega_{0}\rho_{10}\left(t\right)-\nu^{2}\int_{0}^{t}e^{-2\lambda (t-t^{\prime })}e^{-i\omega_{0}\left(t-t^{\prime }\right)}\rho_{10}\left(t^{\prime }\right),\\ \frac{d}{dt}\rho_{01}\left(t\right)= & i\omega_{0}\rho_{01}\left(t\right)-\nu^{2}\int_{0}^{t}e^{-2\lambda (t-t^{\prime })}e^{i\omega_{0}\left(t-t^{\prime }\right)}\rho_{01}\left(t^{\prime }\right). \end{array}$$ By taking the Laplace transform over Equation (Equation 38), the reduced density matrix elements in the Laplace domain can be expressed as
(39)$$\begin{array}{cc} {\tilde{\rho }}_{11}\left(p\right)= & \frac{1}{p}\rho_{11}\left(0\right),\\ {\tilde{\rho }}_{10}\left(p\right)= & \frac{p+i\omega_{0}+2\lambda }{\left(p+i\omega_{0}\right)\left(p+i\omega_{0}+2\lambda \right)+\nu^{2}}\rho_{10}\left(0\right),\\ {\tilde{\rho }}_{01}\left(p\right)= & \frac{p-i\omega_{0}+2\lambda }{\left(p-i\omega_{0}\right)\left(p-i\omega_{0}+2\lambda \right)+\nu^{2}}\rho_{01}\left(0\right),\\ {\tilde{\rho }}_{00}\left(p\right)= & \frac{1}{p}\rho_{00}\left(0\right). \end{array}$$ We can, by means of the inverse Laplace transform of Equation (Equation 39), write the reduced density matrix in time domain as
(40)$$\begin{array}{cc} \rho (t)= & \left(\begin{array}{cc} \rho_{11}\left(0\right) & \rho_{10}\left(0\right)e^{-i\omega_{0}t}F\left(t\right)\\ \rho_{01}\left(0\right)e^{i\omega_{0}t}F\left(t\right) & \rho_{00}\left(0\right) \end{array}\right), \end{array}$$
where F(t) denotes the decoherence factor quantifying the coherence evolution of the quantum system
(41)$$F\left(t\right)=e^{-\lambda t}\left\{\begin{array}{cc} \cosh\left(\beta t\right)+\frac{\lambda }{\beta }\sinh\left(\beta t\right), & \nu <\lambda ,\\ 1+\lambda t, & \nu =\lambda ,\\ \cos\left(\beta t\right)+\frac{\lambda }{\beta }\sin\left(\beta t\right), & \nu >\lambda . \end{array}\right.$$
where $\beta =\sqrt{|\lambda^{2}-\nu^{2}|}$. This expression is consistent with the well-known results obtained in References [57,58,59].

Figure 1 displays the decoherence factor F(t) as a function of the evolution time in the presence of the RTN for different jumping amplitude ν. As depict in the figure, with the increase of the jumping amplitude ν, the behavior in the decoherence factor F(t) displays a transition from monotonic decay to nonmonotonic oscillatory decay: the decoherence factor F(t) decays monotonically for the jumping amplitude ν<λ (weak coupling region), which reflects that the decoherence dynamics of the quantum system is Markovian. The decoherence factor F(t) decays nonmonotonically with coherence revivals for the jumping amplitude ν>λ (strong coupling region), which indicates that the decoherence dynamics of the quantum system becomes non-Markovian. In the weak coupling region, the decay of the decoherence factor F(t) increases with the increase of the jumping amplitude ν, which indicates that the strength of the coupling can enhance the decoherence dynamics of the quantum system. In the strong coupling region, the nonmonotonic oscillations in the decoherence factor become obvious, which reveals that the non-Markovian behavior in the decoherence dynamics of the quantum system is pronounced.

### 3.2. Quantum Dynamics Induced by the OUN Process

We now consider the dynamical evolution of the qubit system when the environmental noise ξ(t) obeys a stationary OUN process with the width γ and decay rate λ of the Gaussian distribution. The time evolution for the conditional probability of the OUN process satisfies the master equation
(42)$$\frac{\partial }{\partial t}P\left(\xi ,t|\xi^{\prime },t^{\prime }\right)=\lambda \frac{\partial }{\partial \xi }\left(\xi +\gamma^{2}\frac{\partial }{\partial \xi }\right)P\left(\xi ,t|\xi^{\prime },t^{\prime }\right),$$
where the initial condition is given by $P\left(\xi ,t^{\prime }|\xi^{\prime },t^{\prime }\right)=\delta \left(\xi -\xi^{\prime }\right)$. The expression of the conditional probability in Equation (Equation 42) can be solved as
(43)$$P\left(\xi ,t|\xi^{\prime },t^{\prime }\right)=\frac{1}{\sqrt{2\pi \gamma^{2}\left[1-e^{-2\lambda (t-t^{\prime })}\right]}}\exp\left\{-\frac{{\left[\xi -\xi^{\prime }e^{-\lambda (t-t^{\prime })}\right]}^{2}}{2\gamma^{2}\left[1-e^{-2\lambda (t-t^{\prime })}\right]}\right\}.$$ For positive λ and large time difference $\lambda (t-t^{\prime })\gg 1$, the stationary distribution of the noise process satisfies
(44)$$P_{\mathrm{st}}\left(\xi \right)=\frac{1}{\sqrt{2\pi \gamma^{2}}}\exp\left(-\frac{\xi^{2}}{2\gamma^{2}}\right).$$ Considering the noise process is Markov, the *n*-point probability distribution can be expressed as
(45)$$P\left(\xi_{n},t_{n};\cdots ;\xi_{1},t_{1}\right)=\prod_{i}^{n-1}P\left(\xi_{i+1},t_{i+1}|\xi_{i},t_{i}\right)P_{\mathrm{st}}\left(\xi_{1}\right),$$

Based on the stationary and Markovian statistical properties of the noise process, the average is zero
(46)〈ξ(t)〉=0, and its correlation function of second-order is exponential
(47)$$\langle \xi \left(t\right)\xi \left(t^{\prime }\right)\rangle =\gamma^{2}e^{-\lambda (t-t^{\prime })}.$$ All the cumulants beyond second order are zero [55,56]
(48)$${\langle \xi \left(t\right)\xi \left(t_{1}\right)\cdots \xi \left(t_{n}\right)\rangle }_{c}=0,$$
for every ordered set of time instants $t>t_{1}>\cdots >t_{n}\left(n\geq 2\right)$. We can, in terms of the statistical characteristics of the environmental noise ξ(t) obtained above, derive that the time-order cumulants beyond second order vanish
(49)$${\langle \xi \left(t\right)\xi \left(t_{1}\right)\cdots \xi \left(t_{n}\right)\rangle }_{\mathrm{oc}}=0.$$

Consequently, we can employ the TCL master equation to describe the time evolution forthe reduced density matrix of the quantum system [36]
(50)$$\begin{array}{cc} \frac{d}{dt}\rho \left(t\right)= & -\frac{i}{2}\left(\omega_{0}L_{\sigma_{z}}+\Delta_{0}L_{\sigma_{x}}\right)\rho \left(t\right)-\frac{1}{4}\int_{0}^{t}dt^{\prime }\langle \xi \left(t\right)\xi \left(t^{\prime }\right)\rangle \\ \; & \times L_{\sigma_{z}}\exp\left[-\frac{i}{2}\left(\omega_{0}L_{\sigma_{z}}+\Delta_{0}L_{\sigma_{x}}\right)\left(t-t^{\prime }\right)\right]L_{\sigma_{z}}\exp\left[\frac{i}{2}\left(\omega_{0}L_{\sigma_{z}}+\Delta_{0}L_{\sigma_{x}}\right)\left(t-t^{\prime }\right)\right]\rho \left(t\right). \end{array}$$ The dynamical evolution of the reduced density matrix elements can be written as
(51)$$\begin{array}{cc} \frac{d}{dt}\rho_{11}\left(t\right)= & \frac{i}{2}\Delta_{0}\left[\rho_{10}\left(t\right)-\rho_{01}\left(t\right)\right],\\ \frac{d}{dt}\rho_{00}\left(t\right)= & -\frac{i}{2}\Delta_{0}\left[\rho_{10}\left(t\right)-\rho_{01}\left(t\right)\right],\\ \frac{d}{dt}\rho_{10}\left(t\right)= & -i\omega_{0}\rho_{10}\left(t\right)+\frac{i}{2}\Delta_{0}\left[\rho_{11}\left(t\right)-\rho_{00}\left(t\right)\right]\\ \; & -\gamma^{2}\int_{0}^{t}e^{-\lambda (t-t^{\prime })}\left\{{\left[\frac{\Delta_{0}^{2}}{2\Omega^{2}}+\frac{\Omega^{2}+\omega_{0}^{2}}{2\Omega^{2}}\cos\Omega \left(t-t^{\prime }\right)\right]}^{2}+\frac{\epsilon_{0}^{2}}{\Omega^{2}}\sin^{2}\Omega \left(t-t^{\prime }\right)\right\}\rho_{10}\left(t\right),\\ \frac{d}{dt}\rho_{01}\left(t\right)= & i\omega_{0}\rho_{01}\left(t\right)-\frac{i}{2}\Delta_{0}\left[\rho_{11}\left(t\right)-\rho_{00}\left(t\right)\right]\\ \; & -\gamma^{2}\int_{0}^{t}e^{-\lambda (t-t^{\prime })}\left\{{\left[\frac{\Delta_{0}^{2}}{2\Omega^{2}}+\frac{\Omega^{2}+\omega_{0}^{2}}{2\Omega^{2}}\cos\Omega \left(t-t^{\prime }\right)\right]}^{2}+\frac{\epsilon_{0}^{2}}{\Omega^{2}}\sin^{2}\Omega \left(t-t^{\prime }\right)\right\}\rho_{01}\left(t\right). \end{array}$$

We consider the uncoupled case $\Delta_{0}=0$. The system undergoes pure decoherence and the time evolution of the reduced density matrix elements in Equation (Equation 51) can be reduced to
(52)$$\begin{array}{cc} \frac{d}{dt}\rho_{11}\left(t\right)= & 0,\\ \frac{d}{dt}\rho_{00}\left(t\right)= & 0,\\ \frac{d}{dt}\rho_{10}\left(t\right)= & -i\omega_{0}\rho_{10}\left(t\right)-\gamma^{2}\int_{0}^{t}e^{-\lambda (t-t^{\prime })}\rho_{10}\left(t\right),\\ \frac{d}{dt}\rho_{01}\left(t\right)= & i\omega_{0}\rho_{01}\left(t\right)-\gamma^{2}\int_{0}^{t}e^{-\lambda (t-t^{\prime })}\rho_{01}\left(t\right). \end{array}$$ We can, by taking the integral over Equation (Equation 52), express the reduced density matrix of the quantum system as
(53)$$\begin{array}{cc} \rho (t)= & \left(\begin{array}{cc} \rho_{11}\left(0\right) & \rho_{10}\left(0\right)e^{-i\omega_{0}t}F\left(t\right)\\ \rho_{01}\left(0\right)e^{i\omega_{0}t}F\left(t\right) & \rho_{00}\left(0\right) \end{array}\right), \end{array}$$
where the decoherence factor F(t) can be written as
(54)$$F\left(t\right)=\exp\left[-\frac{\gamma^{2}}{\lambda^{2}}\left(e^{-\lambda t}-1+\lambda t\right)\right].$$ This expression is compatible with the well-known results obtained in References [20,24].

Figure 2 displays the decoherence factor F(t) as a function of the evolution time induced by the OUN for different width of the distribution γ. The decoherence factor F(t) decays monotonically for arbitrary values of the width of the distribution γ, which indicates that the decoherence dynamics of the quantum system is always Markovian. In addition, as the width of the distribution γ increases, the decay of the decoherence factor F(t) increases, which reveals that the width of the distribution γ can make the decoherence dynamics of the quantum system pronounced.

## 4. Conclusions

We have theoretically studied the dynamics of a quantum system which is coupled to a fluctuating environment within the framework of Kubo-Anderson spectral diffusion. By employing the projection operator technique, we derived the TC and TCL master equations for the dynamical evolution of the quantum system, respectively. Induced by the RTN and OUN, the second order expanded TC and TCL quantum master equations give the exact dynamical evolution of the qubit system, respectively. For the case with no tunneling coupling, the qubit system undergoes pure decoherence. The expressions of the decoherence factor we obtained return to the well-known existing ones. Induced by the RTN, the decoherence dynamics of the quantum system displays a transition from Markovian behavior with monotonic decay to non-Markovian behavior with nonmonotonic oscillatory decay. In the presence of the OUN, the decoherence dynamics of the quantum system decays monotonically and always only shows Markovian behavior. We hope that our investigation will contribute to further understanding of quantum dynamics and will be effective in the suppression and control of the dynamical decoherence of open quantum systems.

## Figures and Tables

**Figure 1 entropy-21-01040-f001:**
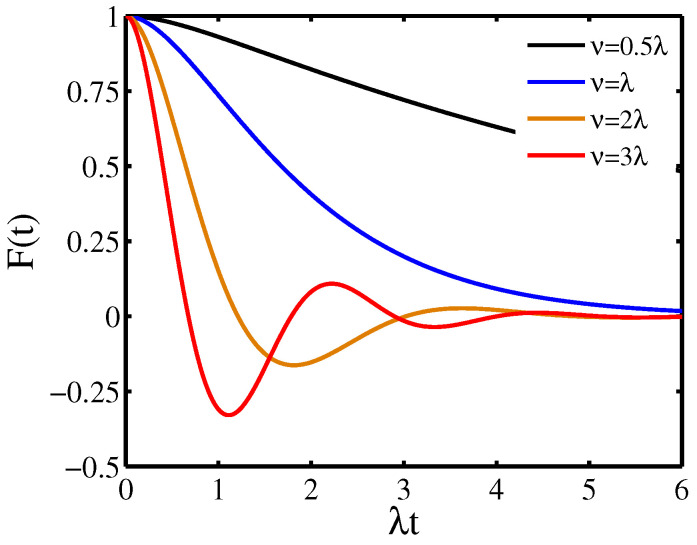
Decoherence factor F(t) as a function of the evolution time induced by the random telegraph noise (RTN) for different jumping amplitude ν.

**Figure 2 entropy-21-01040-f002:**
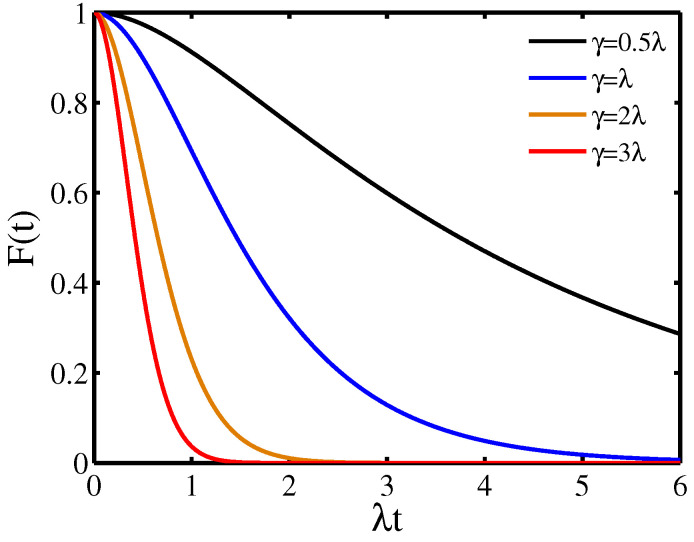
Decoherence factor F(t) as a function of the evolution time in the presence of the Ornstein-Uhlenbeck noise (OUN) for different values of the width of the distribution γ.

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
