# Peer review of "Quantum Dynamics in a Fluctuating Environment"

_entropy, 2019, doi:10.3390/e21111040_

Round 1

Reviewer 1 Report

The manuscript on quantum dynamics in a fluctuating environment develops two types of quantum master equations and applies the equations on a simple two-state qubit system with two types of stochastic noise that include random telegraph noise and Ornstein-Uhlenbeck noise. The manuscript is well written. The projection operator methodology requires a certain level of expertise to fully understand the steps. Fortunately, the author cites many references that will help readers unfamiliar with the methods employed to follow all steps, which are laid out in detail. The author subsequently applies the formal equations on a simple system, and by doing so, verifies that the theoretical development is sound. This is done by demonstrating that the general approach has practical value in the context of solving two special types of noise on a qubit system. The results obtained recover well-known results based on different approaches. My expertise is stretched, so I cannot vouch that the theoretical framework developed in this work is truly novel, nor have I independently verified the correctness of each step in the derivations. However, based on the clarity of author's derivation, with extensive references, and explanations, there appears to me no technical problems and the discussions do not need improvement.  

Author Response

Thank you very much for giving us the positive and constructive comments.

Reviewer 2 Report

The author derives a time convolution-less master equation for a qubit in the presence of classical noise and applies to the case of random telegraph noise and Ornstein Uhlenbeck process. Well known results are reproduced. Relevant literature is not mentioned. For decoherence due to OU noise and to random telegraph noise see also

Rabenstein, K., Sverdlov, V.A. & Averin, D.V. Jetp Lett. (2004) 79: 646. https://doi.org/10.1134/1.1790024;

Paladino E. Faoro L. Falci G. Adv. Sol. State Phys., 43, 747, (2003) 

This article is an exercise which does not bring new physical insight on processes of possible interest in connection with decoherence due to non Markovian environment. I recommend not to publish it.

Author Response

 Thank you very much for your constructive comments. We have added the two references (references [40] and [48] in the revised manuscript) due to their important relevance to the manuscript. 

Reviewer 3 Report

This is very nicely written paper on open quantum system dynamics. I have a few remarks that the author should consider before publication:

There are many different derivations of quantum master equations in the literature using projection operator formalism. The author should make it clear what if anything is new in his presentation of the derivation. The author could expand his discussion on the existing literature on non-Markovian master equations (and include more refs). For example, it has been shown recently that a Lindblad-like equation (but with time-dependent rates) can be derived for ANY quantum system (arxiv 1903.03861 and refs therein). Of course the author does not need to discuss the Lindblad case in particular because it's Markovian. While the projection operator route produces nice formal equations, it is very difficult to use them for correlated quantum systems in general. The two examples given in the paper are nice but somewhat trivial. Could the author comment on using his equations for more complicated systems?

Nevertheless, I can recommend the paper to be published after the author has considered these items.

There are minor grammatical mistakes (e.g. types of master equation -> equations) so one more proofreading is needed.

Author Response

  Thank you very much for giving us the positive and constructive comments. We have added the reference (arxiv 1903.03861) (reference [17] in the revised manuscript) due to its important relevance to the manuscript. We have formally derived two types of quantum master equations for dynamical evolution of the reduced density matrix of the system by employing the projection operator technique which are closely associated with the statistical properties of the environmental noise. However, only for a few simple cases, we can obtain the exact expression of the reduced density matrix of the quantum system based on the two types of quantum master equations we have derived. In most cases, we need to take some approximations to obtain the reduced density matrix of the system or we should know the closure of the higher-order correlation functions of the environmental noise as we have studied in references [33] and [36] in the revised manuscript.

  We have corrected the grammatical mistakes in the manuscript carefully.

  Once again, thank you for your constructive comments and valuable suggestions. We hope sincerely that our replies could meet your approval.